# Delayed diagnosis of congenital cataract in preterm infants: Findings from the IoLunder2 cohort study

**Ameenat Lola Solebo** [1,2,3,4] *, **Jugnoo Sangeeta Rahi** [1,2,3,4], **on behalf of the British Congenital Cataract Interest Group** [¶]

1 Population, Practice and Policy Research and Teaching Department, UCL GOS Institute of Child Health, London, United Kingdom, 2 Ophthalmology Department, Great Ormond Street Hospital NHS Foundation Trust, London, United Kingdom, 3 Ulverscroft Vision Research Group, UCL GOS Institute of Child Health, University College London, London, United Kingdom, 4 Population and Data Sciences, Institute of Ophthalmology and National Institute for Health Research Biomedical Research Centre at Moorfields Eye Hospital NHS Foundation Trust, London, United Kingdom

☺ These authors contributed equally to this work.
¶ Membership of the British Congenital Cataract Interest Group is provided in the Acknowledgments.
* a.solebo@ucl.ac.uk

**Data Availability Statement:** Due to the potentially identifiable nature of the dataset and as participants have not given the necessary consent, the authors are prevented from publicly sharing individual level

## Abstract

### Background and objectives

Early detection is critical to achieving optimal outcomes in children with congenital cataract. We hypothesized that detection of congenital cataract in preterm infants would be delayed compared with term/post-term peers due to delayed delivery of whole population child health interventions.

### Methods

Secondary analysis of data using a nested case-control study approach in a prospective population-based cohort study. Inclusion criteria comprised children diagnosed with congenital cataract requiring surgical intervention during the first two years of life in UK and Ireland in 2009 and 2010. Association between late detection (after eight weeks post-natal age, ie outside the neonatal and infant eye national screening programme) of cataract and preterm birth (gestational age less than 37 weeks) was assessed using multivariable logistic regression.

### Results

Of 186 children with congenital cataract, 17 children were born preterm (9%, gestational age range 24–37weeks). Neonatal detection occurred in 64/186 (34%), and late detection in 64 children (34%). Late detection was independently associated with premature birth, specifically moderate/late preterm birth (adjusted odds ratio 3.0, 95%CI 1.1 to 8.5).

data under the ethics approval for this study from the National Health Service Health Research Authority National Research Ethics Service London – Harrow Committee (Research ethics reference 08/H0714/65). New ethics approval would be needed to either access individual level data without participant consent, or to re-approach participants in order to seek consent for data sharing. For more information, contact harrow.rec@hra.nhs.uk.

**Funding:** AL Solebo received support through an Ulverscroft Vision Research Group fellowship, an Academy of Medical Sciences Lecturer award, from the National Institute for Health Research Biomedical Research Centre (NIHR BRC) based at Moorfields Eye Hospital NHS Foundation Trust and UCL Institute of Ophthalmology, and through an NIHR Clinician Scientist award (CS-2018-18-ST2-005). JS Rahi is supported in part by the NIHR BRC based at Moorfields Eye Hospital NHS Foundation Trust and UCL Institute of Ophthalmology, and an NIHR Senior Investigator award (NF-SI-0617-10031). This work was undertaken at UCL Institute of Child Health / Great Ormond Street Hospital for children which received a proportion of funding from the Department of Health's NIHR Biomedical Research Centres funding scheme. The funders had no role in study design, data collection and analysis, decision to publish, or preparation of the manuscript.

**Competing interests:** The authors have declared that no competing interests exist

## Conclusions

Our findings suggest that, despite enhanced eye surveillance being recommended for those born moderate/late preterm (32+ weeks gestational age, ie not eligible for retinopathy of prematurity screening), congenital cataract is not being effectively detected through the routine screening programme for this vulnerable group. It is necessary to improve the effectiveness of the screening programme, and care must be taken to ensure that competing health care needs of preterm children do not prevent universal child health interventions.

## Introduction

The prevention of childhood blindness is a key public health objective, in recognition of the significant, cumulative and life-long negative impact of early life visual disability, both on the individual and their broader society [1, 2]. The economic impact of childhood blindness has been estimated as at least USD$800,000 per affected child for direct health care costs alone [3]. Globally, cataract is one of the most important preventable causes of childhood blindness [4]. For congenital cataract, treatment (surgery and post operative visual habilitation) within the first few months at life results in the best visual outcomes [4]. Thus, effective treatment requires prompt detection, which also allows prompt diagnosis of associated genetic or other non-ocular disorders. Consequently, universal screening programmes to detect congenital cataract exist in many countries [5–7]. In the United Kingdom (UK) the Newborn and Infant Physical Examination (NIPE) programme comprises clinical examination of the eyes by midwives or paediatricians in the first 72 hours of life alongside hips, heart and testes examination. The eye testing comprises use of the direct ophthalmoscope to assess the red reflex, with congenital cataract as the target eye condition [5]. There is a 'safety net' examination in the community (family practitioners) at age six to eight weeks, coinciding with the first infant immunisations [5]. In the UK, congenital cataract is considered an ophthalmic emergency, necessitating a specialist ophthalmic consultation within two weeks of referral [5].

Between 1 in 13 and 1 in 10 children in high income settings such as the UK and USA are born preterm (ie before 37 weeks of pregnancy) [8] with preterm birth rates, and survival rates following preterm birth, expected to grow globally (including within middle and lower income settings) [8]. Whilst this vulnerable group may have health care needs specific to the complications of their preterm birth, they remain in need of 'whole population' child health care programmes such as vaccinations and neonatal eye screening. Recently, investigators have reported delayed vaccination amongst children born preterm [9, 10]. In this study, we compared the rates of delayed detection of congenital cataract among preterm and term/post term infants diagnosed with visually impactful cataract. We hypothesized that preterm infants would have higher rates of delayed detection compared with term/post-term infants.

## Materials and methods

We undertook secondary analysis of data using a nested case control study approach in a prospective population-based cohort study, the UK and Ireland study of children under 2 years of age undergoing surgery for congenital and infantile cataract (IoLunder2 study) [11].

## Study design and participants

The IOLunder2 study used active surveillance methods and a national clinical research network to identify children undergoing cataract surgery during the first two years of life in UK and Ireland between January 2009 and December 2010.

## Data collection

Following case identification, and written parental consent, detailed clinical and demographic data were collected using standardised study specific collection instruments pre, per and post-operatively, with data collection from recruitment [11]. The prospectively collected standardised dataset for the nationally representative IOLunder2 cohort (with the study having been shown to have ascertained all those children aged under 2 diagnosed with cataract requiring surgical intervention in the UK between January 2009 and December 2010) [12] included data on the referral pathway leading to diagnosis of congenital cataract. Data were also collected on socio-economic deprivation using the standard approach of family residence postcode derived Index of Multiple Deprivation (IMD).

## Analysis

We undertook descriptive analysis of clinical and demographic characteristics. Congenital ocular anomalies were defined as structural malformations affecting the whole globe or specific components (ICD 10 codes Q11, 12, 14 and 15). 'Truly' congenital (rather than possibly infantile onset) cataract was defined as lens opacity co-existent with other congenital structural anomalies (eg persistence of fetal vasculature, anterior segment dysgenesis, whole globe anomalies, or congenital anomalies of lens shape) [13]. Child whose 'truly' congenital cataract was detected at either the neonatal eye health screening check, or at the six to eight week health check was considered to have been detected by the national screening programme [14]. Any child detected after eight weeks of post-natal age was considered to have been detected 'late'.

To assess factors associated with late detection, we used multivariable logistic regression models to estimate adjusted odds ratios (ORs) and 95% CIs. Children with a known history of autosomal dominant inherited cataract, who are recommended to have eye examinations undertaken by ophthalmologists within the first months of life [14], were excluded from these analyses but descriptive subgroup analysis was undertaken to assess if they had undergone the recommended targeted specialist surveillance as a high risk group. We posited that the factors associated with the outcome of late detection of cataract, other than preterm birth, comprised those known to be associated with variations in access to health services, those likely to raise suspicion of cataract, and those associated with additional health care needs, ie non-white ethnicity, socio-economic deprivation status (index of multiple deprivation score within the lowest national quantile), the presence of anterior segment malformations or anomalies involving the whole globe (which may impact on ease of detection of ocular anomaly by non-ophthalmic clinicians) and the presence of a non-ocular impairment or disorder. Correlation between variables was investigated using $\chi 2$ tests (S1 and S2 Tables) adhering to the current conventional threshold of $p<0.05$ for a statistically significant correlation. Multivariable analysis, using backward stepwise regression, included those variables with a p value of $<0.10$ in initial univariable analysis. We retained factors in the multivariable model if they altered the odds ratio estimate by more than 10% or were independently associated at a 5% significance level. Analyses were performed using Stata (version 15.1, Stata Corporation, College Station, Texas, USA).

## Results

The IoLunder2 study recruited 254 children undergoing cataract surgery under the age of two years, of whom 186 (73%) had truly congenital cataract. In 64 of these 186 children (34%) cataract was detected at the NIPE neonatal examination. In 58 children (31%) cataract was detected later than the newborn NIPE examination, but before or at the 6–8 week examination (Table 1).

There were 66 children with bilateral congenital cataract (ie at risk of bilateral blindness) who were not detected at the neonatal exam. Of these 66, 31 had their ocular anomaly detected before or at the 6–8 week examination, and in the remaining children cataract was detected at a median age of 14 weeks (range 10–22 weeks). The median age at detection for the 29 children with unilateral congenital cataract who were not detected by the NIPE screening programme was 22 weeks / 5.1 months (range 9 weeks– 14 months).

The demographic and clinical characteristics of the children is presented in Table 2. Children from a minority ethnic group, resident in areas of socioeconomic deprivation, or who were of premature birth were represented in the cohort. Almost one in five children also had non-ocular abnormalities/conditions (Table 2).

### Timing of detection in children born premature

Within the IoLunder2 cohort, 17 infants were born premature (9% of the cohort), with gestational ages ranging from 24–37 weeks. Of these, 2 were born at less than 32 weeks gestational age and thus would have been eligible for retinopathy of prematurity screening [14] For both these neonates, cataract was detected within the first 3 days of life. The remaining 15 infants were born 'moderate' to 'late' preterm (33 to 37 weeks) [10, 15]. Overall, children born preterm were over-represented amongst those with a late detection of congenital cataract (Table 2). Amongst children born preterm whose cataract was detected late, the median age at detection was 21 weeks, IQR 12–27 weeks, range 10–64 weeks. Following adjustment for gestational age, the median corrected age at detection was 15 weeks, IQR 8–24 (thus for 75% of preterm children cataract was detected later than 8 weeks adjusted age), range 6–60 weeks.

### Timing of detection in those with other disorders

Thirty children had associated non-ocular disorders, including Trisomy 21 (n = 5), metabolic disorders (n = 5), congenital heart defects (n = 8), microcephaly, (n = 3), other congenital structural brain anomalies (n = 5), and hearing impairment (n = 4), with 22 of the 30 children having multiple affected systems. Children with a co-existent non-ocular abnormality were

**Table 1. Number and proportions of children with congenital cataract detected at the different screening stages.**

| | IOLunder2 | | |
|---|---|---|---|
| | **Bilateral cataract** | **Unilateral cataract** | **Total** |
| | **n = 108** | **n = 78** | **n = 186** |
| **Newborn NIPE examination stage** | 42 | 22 | 64 |
| | (39% 95% CI 30–48%) | (28% 19–39%) | (34% 28–42%) |
| **Infant NIPE examination stage*** | 31 | 27 | 58 |
| | (29% 20–38%) | (35% 24–46%) | (31% 25–38%) |
| **After the scheduled NIPE examinations** | 35 | 29 | 64 |
| | (32% 24–41%) | (37% 28–48%) | (34% 28–42%) |

*includes all infants for whom cataract was detected after the neonatal screening window but before or at the 6–8 week NIPE examination

**Table 2. Comparison of demographic, clinical characteristics and visual outcomes by timing of detection.**

| | | IOLunder2 cohort | | |
| --- | --- | --- | --- | --- |
| | | Detected within NIPE | Detected late | Total |
| | | n = 122 | n = 64 | n = 186 |
| Sex | Female (%) | 60 (49%) | 29 (45%) | 89 (48%) |
| Ethnicity | Non-white (%) | 68 (56%) | 39 (61%) | 107 (58%) |
| Deprivation | Living in area in highest scoring quintile for deprivation indices (most deprived) | 30 (25%) | 12 (19%) | 42 (22%) |
| History of consanguinity | Consanguineous parents | 2 (1%) | 0 | 2 (<1%) |
| Multiple birth | Twin or other multiple | 5 (4%) | 4 (6%) | 9 (5%) |
| Family history | Of ocular disorder with or without associated systemic disorder | 37 (30%) | 10* (16%) p<0.05 | 47 (26%) |
| Occurrence of ocular anomalies | Whole globe | 105 (86%) | 46* (72%) p<0.05 | 151 (81%) |
| | Anterior segment dysgenesis | 34 (28%) | 15 (23%) | 49 (26%) |
| Non-ocular disorder / impairment | | 17 (13%) | 13 (20%) | 30 (16%) |
| Prematurity | Born at less than 37 weeks gestational age[&] | 7 (6%) | 10* (16%) p<0.05 | 17 (9%) |
| Visual outcome | Unilateral cataract: Median visual acuity[¶] in operated eye (interquartile range, full range) | 0.9 (0.4–2, 0–3) | 1.3*(0.5–2, 0.1–3) p<0.05 | 1.2 (0.5–2,0–3) |
| | Bilateral cataract: Median visual acuity[¶] both eyes open (IQR, full range) | 0.4(0.3–0.7,0.1–3) | 0.7*(0.4–1, 0.1–3) p<0.05 | 0.6 (0.3–1, 0.1–3) |

*Statistically significant difference between children detected within NIPE and those detected late (χ2 testing)

[¶]Best corrected logarithm of the minimum angle of resolution, logMAR acuity, 0.0 is normal vision, 0.5 moderate impairment, 1.0 is severe visual impairment, 3.0 is no ability to perceive light)

[&]Gestational age data missing for 3 cases, all in the bilateral cataract group. No other missing data

over-represented within the late detection group. Anterior segment anomalies included iris maldevelopment such as hypoplasia or pupillary membranes (n = 32), embryotoxon (abnormalities of the peripheral cornea, n = 12 children), dysgenesis of the anterior segment with adherence of the iris to the cornea (iridocorneal adhesion n = 8). Those with anomalies affecting the whole of the globe were more likely to be in the group detected within the NIPE screening timeline (Table 2). The most common whole globe disorder was microphthalmos, affecting 135 children.

## Timing of detection in those with a known family history

Forty-seven children with congenital cataract were born to families with a history of ocular anomalies. Of these, 17 were known to have autosomal dominant inherited ocular disease, all of whom were detected within the NIPE screening programme. Twenty-five (23 bilaterally affected) of the remaining 30 children with a different family history of congenital cataract, were not detected at their neonatal exam, and 10 children (33%) were detected late.

## Factors associated with late detection of congenital cataract

In univariable analysis, premature birth and the absence of a co-existent ocular anomaly were positively associated with late detection of cataract (Table 3) overall, with differences by laterality of cataract. We identified some correlations amongst explanatory variables: the proportion of children born preterm was higher in those with coexistent systemic disorders than in those children with 'isolated' ocular disorder (S1 Table). Prematurity was also positively associated with family residence within an area of relative deprivation (S1 and S2 Tables). Preterm birth

**Table 3. Univariable logistic regression analysis of factors associated with late detection of all, bilateral and unilateral cataract.**

|  | Odds ratio | 95% CI | P |
|---|---|---|---|
| **All cataract (n = 186)** |  |  |  |
| **Female sex** | 0.86 | 0.47–1.57 | 0.62 |
| **White ethnicity** | 0.81 | 0.4–1.49 | 0.50 |
| **Deprivation** | 0.71 | 0.33–1.49 | 0.37 |
| **Prematurity** | 3.04 | **1.10–8.42** | **0.03** |
| **Non ophthalmic disorder** | 1.57 | 0.71–3.49 | 0.26 |
| **Ant segment dysgenesis** | 0.79 | 0.39–1.59 | **0.52** |
| **Whole globe anomaly** | 0.41 | 0.19–8.74 | **0.02** |
| **Bilateral cataract (n = 108)** |  |  |  |
| **Female sex** | 1.52 | 0.68–3.42 | 0.31 |
| **White ethnicity** | 0.96 | 0.43–2.17 | 0.92 |
| **Deprivation** | 0.7 | 0.29–1.74 | 0.45 |
| **Prematurity** | 3.3 | 1.05–10.44 | **0.04** |
| **Non-ocular disorder** | 1.4 | 0.57–3.57 | 0.45 |
| **Anterior segment dysgenesis** | 1.60 | 0.61–4.19 | 0.34 |
| **Whole globe anomaly** | 0.36 | 0.10–1.06 | **0.10** |
| **Unilateral cataract (n = 78)** |  |  |  |
| **Female sex** | 0.39 | 0.15–1.13 | 0.16 |
| **White ethnicity** | 0.65 | 0.25–1.67 | 0.36 |
| **Deprivation** | 0.83 | 0.19–3.59 | 0.80 |
| **Prematurity** | 3.56 | 0.31–41.05 | 0.31 |
| **Non-ocular disorder** | 5.53 | 0.55–55.96 | 0.15 |
| **Anterior segment dysgenesis** | 0.35 | 0.12–1.00 | **0.05** |
| **Whole globe anomaly** | 0.46 | 0.17–1.23 | **0.10** |

was the strongest independent predictor of the risk of late detection for those with bilateral cataract, and for all children with cataract (Table 4).

## Discussion

In a population-based cohort study of children requiring surgery for cataract, we found a third of those with truly congenital disease (ie present at birth), were not detected via the national screening programme to detect cataract. Preterm birth was independently associated with late detection.

**Table 4. Multivariable regression analysis of factors independently associated with late detection of cataract.**

|  | Adjusted odds ratio* | 95% CI | p |
|---|---|---|---|
| **All cataract (n-186)** |  |  |  |
| **Prematurity** | 3.01 | 1.07–8.45 | **0.04** |
| **Whole globe anomaly** | 0.42 | 0.20–0.89 | **0.02** |
| **Bilateral cataract (n = 108)** |  |  |  |
| **Prematurity** | 3.02 | 1.00–9.73 | **0.04** |
| **Whole globe anomaly** | 0.41 | 0.11–1.50 | 0.18 |
| **Unilateral cataract (n = 78)** |  |  |  |
| **Ant segment dysgenesis** | 0.32 | 0.11–0.96 | **0.04** |
| **Whole globe anomaly** | 0.41 | 0.14–1.16 | 0.10 |

This study presents a UK population level 'snapshot' of those children undergoing cataract surgery in infancy, with very high levels of ascertainment. Other findings from the IoLunder2 cohort on outcomes after surgery and the determinants of outcome have since been replicated by investigators from other similar disease cohorts [16, 17], supporting the external validity of our study findings. IoLunder2 is missing data on detection pathways for children with visually insignificant cataract (ie where surgery was not necessary), or those with severe ocular anomaly or non-ocular disease where visual prognosis was too poor for other reasons to support surgical intervention. These children are, however, a minority of all those diagnosed with congenital cataract [6, 17–19], and our study remains able to report on the predictors of late detection for those children in whom timely detection is imperative to good outcomes following surgery. Another possible limitation of the study is that it captures outcomes from the neonatal screening programme as implemented more than a decade ago. However, screening processes have not changed since this study was undertaken [5]. The data reported here are otherwise unavailable. As in many other countries, a public health notification system for congenital anomalies (the National Congenital Anomaly System, NCAS) exists but is limited by low ascertainment of ocular anomaly and the absence of necessary historical clinic data on specific ocular conditions [18]. The IoLunder2 dataset remains the only nationally representative dataset with which to examine the factors affecting the detection of visually significant congenital and infantile cataract.

Premature birth is recognised as a significant adverse event with regards to later life neurodevelopmental outcome, with children's vision affected through insult to, or poor development of cortical visual pathways, with preterm children also being more likely to have common childhood ocular disorders such as strabismus (squint) or refractive errors (focusing) [15, 19, 20]. Children born significantly premature (ie those born under 32 weeks gestation or less than 1501g birthweight) undergo neonatal eye examinations in order to screen for the presence of retinopathy of prematurity, another important preventable cause, globally, of childhood visual disability [21]. Our findings show that cataract is successfully detected amongst these children as ophthalmologists are screening them for ROP. However those children born at a later preterm age (ie, between 32 and 37 gestation), are recommended to undergo the 'normal' neonatal health screening tests and examinations [22, 23]. The reasons why this is not occurring are not clear, and include the possibility that the test as performed is less effective for preterm children, or that the test is not being performed for these children within the recommended schedule.

The test used within a screening programme underpins the viability, effectiveness and implementation of that programme. For newborn and infant eye screening, that test is the clinical ophthalmoscopic examination of the ocular red reflex [5], and external inspection of the eye. This test can be challenging for the non-ophthalmic health professionals who usually perform it (in the UK this typically comprises neonatologists, midwives and family practitioners) [5] and has high variability in reported sensitivity and specificity rates across different health care settings [5]. Without progress in either developing a better test and/or better equipping clinicians to undertake the test with greater accuracy, the scope for improving timely detection may remain limited. There is cause to be optimistic that innovations in infra-red imaging, smart phone based applications, and automated analysis of acquired images [24, 25] may address this problem.

Investigations have reported under- or delayed vaccination of preterm infants versus their term-born peers–ie, challenges delivering necessary whole population interventions to these children within the recommended schedule [10, 26]. The 'healthy child' population health care pathways may not be fully implemented for infants who are often already managed by specialised and or multi-disciplinary teams for other disorders and vulnerabilities. Competing health

needs may require prioritisation by professionals or families of more pressing clinical concerns over the routine 'healthy-child' consultations, or children may have a concomitant period of illness or hospitalisation that prevents their scheduled screening / routine assessment. There may also be suboptimal co-ordination of care provided by a complex mixture of specialist paediatric and community (ie preventative medicine) teams. There is increasing recognition of the additional health needs of those born 'late preterm' [27], and the population of preterm infants continues to rise, particularly in lower and middle income countries, due to increasing incidence of premature birth and improved infant survival [28]. Health professionals involved in the care of preterm children should be alert to the risks to delivery of the 'core' pathway of the public health interventions aimed at improving child visual health through secondary prevention of disease and disability, and support the education of families to counsel them of the need to attend routine community based visits alongside specialist care appointments. The targeted eye screening of the growing global population of premature and low birth weight babies at risk of retinopathy remains important [29], but should be considered alongside the preventative health needs of the growing population of those born mid to late preterm. is particularly important at a time when socioeconomic vulnerability, a key predictor of preterm birth, is expected to worsen globally due to the economic and health care disruption caused by the COVID-19 pandemic [28].

Our findings identify important scope for improvement in the delivery of the neonatal eye screening programme, and possible necessary improvements in other routine or recommended early life public health processes for children born preterm. Attention is warranted to the training of non-expert examiners undertaking a challenging clinical examination. Improved communication of the findings of diagnostic examinations to referring bodies is needed, as well as improved data reporting systems for the outcome of screening processes, in order to create a closed audit loop. Further is needed on the clinical effectiveness of innovative screening tools for congenital cataract, and the child, family and care structure or system level challenges and opportunities for the implementation of whole population child health interventions aimed at improving visual outcomes for children born preterm.

## Supporting information

**S1 Table. Investigation of correlations between variables considered in analysis of outcome–bilateral cataract.**
(DOCX)

**S2 Table. Investigation of correlations between variables considered in analysis of outcome–unilateral cataract.**
(DOCX)

**S1 Checklist. *PLOS ONE* clinical studies checklist.**
(DOCX)

**S2 Checklist. STROBE statement—checklist of items that should be included in reports of observational studies.**
(DOCX)

## Acknowledgments

Membership of the British Isles Congenital Cataract Interest Group:

Lead: Prof Jugnoo S Rahi, j.rahi@ucl.ac.uk. Members: Mr J Abbott, Mr M Parulekar, Mr J Ainsworth, Birmingham Children's Hospital; Miss GW Adams, Professor P Khaw, Ms M

Theodorou, Ms J Hancox, Ms A Dahlmann-Noor, Moorfields Eye Hospital; Ms L Allen, Addenbrokes Hospital; Mr L Amaya, St Thomas' Hospital; Ms S Anwar, Leicester Royal Infirmary; Ms J Ashworth, Mr S Biswas, Manchester Royal Eye Hospital; Mr J Barry, Birmingham and Midland Eye Centre; Professor P Bloom, Western Eye Hospital; Mr R Bowman, Ms I Russell-Eggitt, Mr W Moore, Prof AT Moore, Great Ormond Street Children's Hospital; Mr J Bradbury, Ms R Pilling, Mr T Gout, Bradford Royal Infirmary; Mr D Brosnahan, Our Lady's Children Hospital Dublin; Mr J Butcher, Countess of Chester Hospital; Mr TKJ Chan, Sheffield Children's Hospital; Mr Arvind Chandna, Alder Hey Children's Hospital; Ms J Choi, Sheffield Children's Hospital; Ms AJ Churchill, Bristol Eye Hospital; Mr J Clarke, James Cook University Hospital; Mr MP Clarke, Mr A Shafiq, Royal Victoria Infirmary; Ms F Dean, University Hospitals Coventry; Professor G Dutton, Yorkhill Hospital; Mr J Elston, Oxford Eye Hospital; Mr J Ferris, Cheltenham General Hospital, Dr B Fleck, Princess Alexandra Eye Pavilion; Mr ND George, Ninewells Hospital; Mr L Gnanaraj, Sunderland Eye Infirmary; Mr R Gregson, Nottingham University Hospital; Mr P Hodgkins, Southampton General Hospital; Mr D Jones, Royal Cornwall Hospital; Ms A Joseph, North Staffordshire University Hospital; Mr D Laws, Singleton Hospital; Mr T Lavy, Yorkhill Hospital; Prof C Lloyd, Manchester Royal Eye Hospital and Great Ormond Street Hospital; Mr V Long, Leeds General Infirmary; Dr M MacCrae, Princess Alexandra Eye Hospital; Ms J MacKinnon, Yorkhill Hospital, Mr R Markham, Bristol Eye Hospital; Ms J Marr, Sheffield Children's Hospital; Ms K May, Mr J Self, Southampton Royal Eye Unit; Ms E Mc Loone, Mr G McGinnity, Royal Victoria Hospital Belfast; Dr A Mulvihill, Princess Alexandra Eye Pavilion; Mr W Newman, Manchester Royal Eye Hospital; Mr Q Mansor, South Tees NHS Trust; Mr H Porooshani, Midd Essex Hospital; Mr J Pauw, Clacton Hospital; Mr N Puvanachandra, Norfolk and Norwich Hospitals; Mr AG Quinn, Royal Devon and Exeter Hospital; Dr C Roberts, Western Eye & Imperial Hospitals; Mr CS Scott, Royal Aberdeen Hospital; Mr H Soeldner, James Cook University Hospital; Ms T Sleep, Torbay Hospital; Professor DSI Taylor, Institute of Ophthalmology; Mr R H Taylor, York Hospitals NHS Foundation Trust; Mr P Watts, Cardiff Eye Unit; Mr W Aclimandos, Kings College Hospital; Mr A Aguirre Vila-Coro, Huddersfield Royal Infirmary; Ms C Williams, Bristol Eye Hospital; Mr A Vivian, West Suffolk Hospital; Mr G Woodruff, Leicester Royal Infirmary; Mr S Aftab, Scunthorpe General Hospital, Mr L Amanat, James Paget Hospital; Mr S Armstrong, Countess of Chester Hospital; Mr A Assaf, Milton Keynes Hospital; Mr N Astbury, West Norwich Hospital; Mr Bates, Pembury Hospital, Mr A Beckingsale, Essex County Hospital; Mr G Bedford, Dumfries & Galloway Royal Infirmary; Ms N Boyle, Kilmarnock Hospital; Mr L Benjamin, Stoke Mandeville Hospital; Miss B Billington, Royal Berkshire Hospital; Mr A Blaikie, Queen Margaret Hospital; Miss T Blamires, Northampton General Hospital; Mr D Boase, Queen Alexandra Hospital; Miss M Boodhoo, St Peter's Hospital; Mr J Brazier, Middlesex Hospital; Professor A Bron, Oxford Eye Hospital; Mr R Brown, North Staffordshire University Hospital; Mr I Brown, Old Rectory; Mr S Bryan, Whipps Cross Hospital; Miss P Burgess, Princess Margaret Hospital; Mr J Burke, Royal Hallamshire Hospital; Miss L Butler, Birmingham & Midland Eye Centre; Mr D Calver, Guy's Hospital; Mr A Casswell, Sussex Eye Hospital; Mr M Cole, Torbay Hospital; Mr R Condon, St Peter's Hospital; Mr P Corridan, Wolverhampton Eye Infirmary; Mr R Darvell, Kent and Canterbury Hospital; Mr B Das, Alexandra Hospital; Mr S Daya, Queen Victoria Hospital; Mr R De Cock, Kent and Canterbury Hospital; Mr C Dees, James Cook University Hospital; Mr C Edelsten, Ipswich Hospital; Mr R Edwards, Kent and Canterbury Hospital; Mr H El-Kabasy, Southend Hospital; Mr A Evans, Queen Alexandra Hospital; Mr N Evans, Royal Eye Infirmary; Miss D Flaye, Herts and Essex Hospital; Dr A Gaskell, Ayr Hospital; Miss M Gibbens, Queen Mary's Hospital; Mr C Gibbons, North Devon District Hospital; Mr P Gregory, Conquest Hospital; Mr J Hakim, Queen Mary's Hospital; Mr S Hardman-Lea, Ipswich Hospital; Mr M Hassan, Barnsley

District Hospital; Mr M Heravi, William Harvey Hospital; Ms M Hingorani, Bedford Hospital; Mr R Holden, Derbyshire Royal Infirmary; Mr R Humphrey, Odstock Hospital; Mr C Hutchinson, Royal Halifax Infirmary; Mr J Innes, Hull Royal Infirmary; Mr I Jalili, Roman Bank; Mr C Jenkins, Kent County Ophthalmic and Aural Hospital; Dr E Johnson, Gloucestershire Royal Hospital; Mrs N Kayali, Whipps Cross Hospital; Mr S Keightley, North Hampshire Hospital; Mr P Kinnear, Charing Cross Hospital; Mr A Kostakis, Doncaster Royal Infirmary; Mr S Kotta, Grimsby District General Hospital; Mr R Kumar, Coventry and Warwickshire Hospital; Ms J Leitch, Sutton Hospital; Mr C Liu, Sussex Eye Hospital; Ms C MacEwen, Ninewells Hospital; Mr G Mackintosh, Yew Tree House; Mr A Mandal, Barnsley District Hospital; Mr J McConnell, Ferrers; Mr B McLeod, Sussex Eye Hospital; Mr B Moriarty, Leighton Hospital; Dr G Morrice, Stirling Royal Infirmary; Mr R Morris, Southampton Eye Hospital; Mr N Neugebager, Leighton Hospital; Mr J Nolan, University College Hospital; Mr G O'Connor, Cork University Hospital; Miss R Ohri, Whipps Cross Hospital; Mr S Perry, Kidderminster General Hospital; Mr R Phillips, Arrowe Park Hospital; Dr B Power, Dumfries and Galloway Royal Infirmary; Mr N Price, Cheltenham General Hospital; Mr I Qureshi, Birch Hill Hospital; Mr A Rahman, Pilgrim Hospital; Mr A Reddy, Royal Aberdeen Children's Hospital; Mr E Rosen, St John Street; Mr S Scotcher, Hereford Hospital; Mr J Scott, Stirling Royal Infirmary; Mr P Sellar, West Cumberland Hospital; Mr A Shun Shin, Wolverhampton Eye Infirmary; Mr P Simcock, Royal Devon and Exeter Hospital; Mr I Simmons, St James' University Hospital; Mr JD Stokes, Dublin; Mr M Tappin, St Peter's Hospital; Mr V Thaller, Royal Eye Infirmary; Mr M Thoung, Broomfield Hospital; Mr W Tormey, Waterford Regional Hospital, Mr S Tuft, Moorfields Eye Hospital; Mr M Tutton, Countess of Chester Hospital; Mr J Twomey, Musgrove Park Hospital; Mr S Verghese, West Cumberland Hospital; Ms S Vickers, Sussex Eye Hospital; Mr G Wright, Burnley General Hospital, and Ms AL Solebo, Great Ormond Street Hospital.

## Author Contributions

**Conceptualization:** Ameenat Lola Solebo, Jugnoo Sangeeta Rahi.

**Data curation:** Ameenat Lola Solebo.

**Formal analysis:** Ameenat Lola Solebo.

**Funding acquisition:** Ameenat Lola Solebo, Jugnoo Sangeeta Rahi.

**Investigation:** Ameenat Lola Solebo, Jugnoo Sangeeta Rahi.

**Methodology:** Ameenat Lola Solebo, Jugnoo Sangeeta Rahi.

**Project administration:** Ameenat Lola Solebo.

**Resources:** Ameenat Lola Solebo.

**Software:** Ameenat Lola Solebo.

**Supervision:** Ameenat Lola Solebo, Jugnoo Sangeeta Rahi.

**Validation:** Ameenat Lola Solebo.

**Visualization:** Ameenat Lola Solebo.

**Writing – original draft:** Ameenat Lola Solebo.

**Writing – review & editing:** Ameenat Lola Solebo, Jugnoo Sangeeta Rahi.

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
