## [Decision Letter · Decision Letter 0]

14 Mar 2023

PONE-D-23-01395Delayed diagnosis of congenital cataract in preterm children: findings from the IoLunder2 cohort studyPLOS ONE

Dear Dr. Solebo,

Thank you for submitting your manuscript to PLOS ONE. After careful consideration, we feel that it has merit but does not fully meet PLOS ONE’s publication criteria as it currently stands. Therefore, we invite you to submit a revised version of the manuscript that addresses the points raised during the review process.

ACADEMIC EDITOR:While the manuscript presents important information, the concerns of the reviewers about the accuracy of reporting the information merit attention. Please address all reviewer comments. ==============================

We look forward to receiving your revised manuscript.

Kind regards,

Nader Hussien Lotfy Bayoumi, M.D., FRCS (Glasgow)

Academic Editor

PLOS ONE

4. One of the noted authors is a group or consortium “British Congenital Cataract Interest Group”. In addition to naming the author group, please list the individual authors and affiliations within this group in the acknowledgments section of your manuscript. Please also indicate clearly a lead author for this group along with a contact email address.’

Reviewers' comments:

Reviewer's Responses to Questions

**Comments to the Author**

1. Is the manuscript technically sound, and do the data support the conclusions?

Reviewer #1: Yes

Reviewer #2: Yes

2. Has the statistical analysis been performed appropriately and rigorously? 

Reviewer #1: Yes

Reviewer #2: No

3. Have the authors made all data underlying the findings in their manuscript fully available?

Reviewer #1: No

Reviewer #2: Yes

4. Is the manuscript presented in an intelligible fashion and written in standard English?

Reviewer #1: No

Reviewer #2: Yes

5. Review Comments to the Author

Reviewer #1: Dear authors

I would like to thankyou for the great effort done in preparing your manuscript entitled (Delayed diagnosis of congenital cataract in preterm children: findings from the

IoLunder2 cohort study).

I believe the outcome of the current study will focus the light on an important health problem affecting the late preterm infants.

However, I had some questions and comments hat I embeded in the uploaded PDF file. I hope these points will help you to make the manuscript better.

In addition,you stated in the manuscript that (Individual participant data will not be made available for this rare disease cohort study).On the other hand the journal policy states that (The PLOS Data policy requires authors to make all data underlying the findings described in their manuscript fully available without restriction, with rare exception). They asked me if (the authors have made all data underlying the findings in their manuscript fully available) and I answered (NO).

My best regards

Reviewer #2: The external validity of the research findings is questionable in the current research as it couldn’t be generalized to the population currently . the data obtained from 2010 surveillance. The situation might be totally differe nowadays.

The sample size in the current study is not justifiable as it include only those retrived by the surviellance system (the study might be underpowered)

The authors mentioned that they retain variables in the regression model with a P value <0.10 (is their any reference for that?)

In the tables, the percentage is aways approximated (the total is not 100%) in all tables

Table 1 is highly misleading and difficult to be understood also the median age at the time of diagnosis is not seen in the table however it is important (we can use in this table the terms delayed or timely detection)

Table 2: no need for a row to represeent missing cases (we can add if any as a footnote)

Table 2 doesn’t show a test of significance to reveal the difference between groups (where are the results of bivariate analysis). “minority ethnic group, residence in areas of socioeconomic, deprivation, premature birth or significant systemic diagnoses or impairments were common findings”……this is totally unaccepted . we rely on hypothesis testing”. So you need to add that there is no statistically significant difference……morover at the p value column to the table for better visualizing the results

Did you exclude those with missing gestational age from the total or included (in table 2 ,item 9)

Table 2 includes much information, it can be divided ito 2 tables. Also the visual acuity is not a risk factor for delayed detection, it is an outcome, I think it should be better presented even in a separate table with a clear test of significance. In the same point ,u can separate the simple range from the interquartile to be better understood)

Table 3: what is meant by univariate regression analysis, I think this is a mistake .this is just univariate analysis (using chi square test with estimation of risk)not regression analysis???

authors can combine the data in table 2 and 3 as stratification into unilateral and bilateral cataract didn’t add a lot.

In table 4, the multivariate analysis ,just add the significance of the whole model and overall significance

No comparisons in the discussion between results of the current work and other similar studies. moreover, no mentioned limitations for the presented work

6. PLOS authors have the option to publish the peer review history of their article (what does this mean?). If published, this will include your full peer review and any attached files.

Reviewer #1: No

Reviewer #2: No

---

## [Author Response · Author response to Decision Letter 0]

17 May 2023

Many thanks - a point by point response has been submitted, and is pasted below 

Reviewer #1 

“I believe the outcome of the current study will focus the light on an important health problem affecting the late preterm infants.” We thank the reviewer for their comments 

Line 81 

The sentence needs revision Reworded Line 87-89

Line 85 

This sentence needs revision This has been amended 91-95

Line 170 

The sentence does not give a clear meaning. Please make it more comprehensive. This has been amended 186 - 191

Line 171 

The written results do not match with Table 1 We have rewritten this section for clarity 186 - 191

Line 171 to 173

The sentence needs revision to be more comprehensive to the reader This has been amended 186- 191

Table 1 

I believe the table heading needs revision. The table summarizes the numbers and percentages of infants with congenital cataract detected at different phases Amended to Number and proportions of children with congenital cataract detected at the different screening stages Table 3

Table 1 

I don't believe that there is a value to calculate the 95% confidence interval here. We disagree: it allows the reader a better understanding of the likely uncertainty around the reported proportions from different groups No changes made 

Line 180 

"children" is a very general wrd, please make it more specific Changed to ‘infants’ Line 244

Line 183 

the variable was described as (Multi-system disorder / impairment) in Table 2, please try to use the same term in the text

 For clarity we have used the term ‘non-ocular disorder / impairment’ Text

Table 2

Additional information about the commonly associated ocular abnormalities should be mentioned if available

 These have been provided Lines 233 – 238 

Reviewer #2 

The external validity of the research findings is questionable in the current research as it couldn’t be generalized to the population currently . the data obtained from 2010 surveillance. The situation might be totally differe nowadays.

 We disagree that our findings have limited applicability now as it seems unlikely that the situation regarding screening could be totally different now, other than the situation being considerably worse due to the negative impact of the pandemic on the delivery of preventative medicine globally. 

Within the UK, the NIPE programme has been reviewed twice in the past decade (the authors are advisers to the UK National Screening Committee on the eye component of the NIPE) and no aspect of the NIPE programme content re eyes has changed ie timing, training of staff undertaking the examinations or content of the examination since 2010. We are not aware of any changes to existing mature screening programmes in other countries which would impede the generalisability of our findings to those settings. No change

: The sample size in the current study is not justifiable as it include only those retrived by the surviellance system (the study might be underpowered) We thank the reviewer for giving us chance to describe the power of the IoLunder2 cohort as a nationally representative dataset. As we set out in lines 126 – 128, and as we had reported in an earlier paper (Solebo et al, ref 12 in this manuscript) our active surveillance approach ascertained all cases of cataract surgery in a child under 2 years within the surveillance window in the country. This was shown through comparison with the National Health Services’ data warehouse. No change

The authors mentioned that they retain variables in the regression model with a P value <0.10 (is their any reference for that?) Thank you for the comment. It is routine practice to use this approach to building a multivariable model (for example see https://www.stata.com/manuals/rstepwise.pdf) 

We have not referenced this as we do not consider it necessary, just as we have not have referenced the p value threshold of 0.05 for statistical significance. 

 No change

In the tables, the percentage is aways approximated (the total is not 100%) in all tables We have avoided the use of decimal points to aid legibility. No change

Table 1 is highly misleading and difficult to be understood also the median age at the time of diagnosis is not seen in the table however it is important (we can use in this table the terms delayed or timely detection) We note the confusion caused by table 1 and have amended this for clarity Table 1

Table 2: no need for a row to represeent missing cases (we can add if any as a footnote)

 Amended Table 2

Table 2 doesn’t show a test of significance to reveal the difference between groups (where are the results of bivariate analysis). “minority ethnic group, residence in areas of socioeconomic, deprivation, premature birth or significant systemic diagnoses or impairments were common findings”……this is totally unaccepted . we rely on hypothesis testing”. So you need to add that there is no statistically significant difference……morover at the p value column to the table for better visualizing the results

 The reviewer has we think assumed that this sentence on patient characteristics being common referred to them being MORE common in one group versus another. The sentence to which the reviewer refers is a descriptive analysis statement: hypothesis testing is not appropriate here. In order to avoid confusion we have stated that children with these characteristics are ‘represented’ in our cohort. Line 203 – 205 

Did you exclude those with missing gestational age from the total or included (in table 2 ,item 9) We have been explicit about the extent of missing data on gestational age in the total Table 2

Table 2 includes much information, it can be divided ito 2 tables. Also the visual acuity is not a risk factor for delayed detection, it is an outcome, I think it should be better presented even in a separate table with a clear test of significance. In the same point ,u can separate the simple range from the interquartile to be better understood)

 We suggest that the flow of the article is better served by having these patient characteristics all presented in one table. It will, we think, be clear to the reader that we are not suggesting that all of these characteristics are being considered as risk factors No change

Table 3: what is meant by univariate regression analysis, I think this is a mistake .this is just univariate analysis (using chi square test with estimation of risk)not regression analysis??? Univariable logistic regression is the standard starting point in any statistical analysis that proceeds to multivariable analysis - as described for example in Wallisch C et al. (2022) Review of guidance papers on regression modeling in statistical series of medical journals. PLoS ONE 17(1): e0262918. https://doi.org/10.1371/journal.pone.0262918) 

No change

authors can combine the data in table 2 and 3 as stratification into unilateral and bilateral cataract didn’t add a lot. We have continued to keep these tables separate as there is clinical importance to meaningful differences between detection of bilateral and unilateral cataract. Firstly it has been suggested that unilateral cataract may be easier to detect as the examiner has a healthy contralateral eye for comparison. Secondly differences in outcomes between unilateral and bilateral cataract are well established. No change

In table 4, the multivariate analysis ,just add the significance of the whole model and overall significance We think that the reviewer is referring here to the multivariable analysis we have undertaken. We have taken the appropriate approach to reporting the findings from a multivariable regression model. No change

No comparisons in the discussion between results of the current work and other similar studies. We welcome any suggestions the reviewer may have on citable work on our hypothesis (that preterm infants have higher rates of delayed cataract detection compared with term/post-term infants). We have struggled to find similar studies and have concluded that our findings are novel. We have however contextualised our findings using work from other teams on delays in delivering other public health interventions, specifically vaccination, in preterm infants. These are discussed in the manuscript (references 9 and 10). No change 

moreover, no mentioned limitations for the presented work The reviewer has missed our discussion of limitations in the second paragraph of the paper Line 276 - 288

---

## [Decision Letter · Decision Letter 1]

12 Jun 2023

Delayed diagnosis of congenital cataract in preterm infants: findings from the IoLunder2 cohort study

PONE-D-23-01395R1

Dear Dr. Solebo,

We’re pleased to inform you that your manuscript has been judged scientifically suitable for publication and will be formally accepted for publication once it meets all outstanding technical requirements.

Kind regards,

Nader Hussien Lotfy Bayoumi, M.D., FRCS (Glasgow)

Academic Editor

PLOS ONE

Additional Editor Comments (optional):

Reviewers' comments:

Reviewer's Responses to Questions

**Comments to the Author**

1. If the authors have adequately addressed your comments raised in a previous round of review and you feel that this manuscript is now acceptable for publication, you may indicate that here to bypass the “Comments to the Author” section, enter your conflict of interest statement in the “Confidential to Editor” section, and submit your "Accept" recommendation.

Reviewer #1: All comments have been addressed

2. Is the manuscript technically sound, and do the data support the conclusions?

Reviewer #1: Yes

3. Has the statistical analysis been performed appropriately and rigorously? 

Reviewer #1: Yes

4. Have the authors made all data underlying the findings in their manuscript fully available?

Reviewer #1: Yes

5. Is the manuscript presented in an intelligible fashion and written in standard English?

Reviewer #1: Yes

6. Review Comments to the Author

Reviewer #1: I would like to thank the authors for their great work. They responded very well to all the raised questions and did all the required amendments.

7. PLOS authors have the option to publish the peer review history of their article (what does this mean?). If published, this will include your full peer review and any attached files.

Reviewer #1: **Yes: **Islam SH Ahmed

---

## [Editor Report · Acceptance letter]

10 Aug 2023

PONE-D-23-01395R1 

Delayed diagnosis of congenital cataract in preterm infants: findings from the IoLunder2 cohort study 

Dear Dr. Solebo:

I'm pleased to inform you that your manuscript has been deemed suitable for publication in PLOS ONE. Congratulations! Your manuscript is now with our production department. 

Kind regards, 

on behalf of

Professor Nader Hussien Lotfy Bayoumi 

Academic Editor

PLOS ONE